# DeblurSDI: Blind Image Deblurring Using Self-diffusion

## Abstract

Blind image deconvolution is a challenging ill-posed inverse problem, where both
the latent sharp image and the blur kernel are unknown. Traditional methods often
rely on handcrafted priors, while modern deep learning approaches typically re-
quire extensive pre-training on large external datasets, limiting their adaptability
to real-world scenarios. In this work, we propose DeblurSDI, a zero-shot, self-
supervised framework based on self-diffusion (SDI) that requires no prior training.
DeblurSDI formulates blind deconvolution as an iterative reverse self-diffusion
process that starts from pure noise and progressively refines the solution. At each
step, two randomly-initialized neural networks are optimized continuously to re-
fine the sharp image and the blur kernel. The optimization is guided by an ob-
jective function combining data consistency with a sparsity-promoting $\ell_1$-norm
for the kernel. A key innovation is our noise scheduling mechanism, which stabi-
lizes the optimization and provides remarkable robustness to variations in blur ker-
nel size. These allow DeblurSDI to dynamically learn an instance-specific prior
tailored to the input image. Extensive experiments demonstrate that DeblurSDI
consistently achieves superior performance, recovering sharp images and accurate
kernels even in highly degraded scenarios.

## 1 Introduction

Image deblurring is a fundamental task in computer vision, aiming to recover a sharp image from a
blurred observation. The blur can arise from various sources, such as camera shake, object motion,
or defocus, whose degradation model is typically formulated with the convolution between the latent
sharp image and the blur kernel. In blind image deconvolution, both image and kernel are unknown
and must be jointly estimated from a blurry image alone, which is a particularly challenging ill-
pose inverse problem. Traditional blind deblurring methods rely on handcrafted priors, such as
gradient sparsity (Fergus et al., 2006) or variational Bayesian frameworks (Levin et al., 2007), and
are effective for simple blur scenarios but can be sensitive to complex blurs and noise. In contrast,
modern deep learning approaches leverage large-scale datasets to learn complex mappings from
blurred to sharp images (Nah et al., 2017; Kupyn et al., 2018; Laroche et al., 2024). However,
these methods often require extensive pre-training, which limits their generalization to new blur
types or real-world conditions with varying blurring conditions. More recently, a different line of
research has emerged that applies the principles of Deep Image Prior (DIP) to this task (Ren et al.,
2020). While these methods optimize a randomly-initialized network to fit a single observed image,
leveraging its implicit structure as a regularizer without external training data, they are often sensitive
to the joint optimization process and require careful adjustment of parameters, such as the kernel
size, which poses a significant challenge for practical applications.

In this work, we introduce DeblurSDI, a novel framework that uniquely addresses the challenges of
blind image deblurring. Our primary contribution is the reformulation of this ill-posed problem as a
zero-shot, reverse self-diffusion process, which allows for dynamic, instance-specific optimization
without reliance on large training datasets. To overcome the inherent instability of jointly estimating
the image and blur kernel, we introduce a novel noise scheduler for both the image and kernel prior
networks. This design is central to our method's success, providing exceptional robustness to varia-
tions in blur kernel size—a significant limitation of prior methods. Through extensive experiments,
we demonstrate that DeblurSDI consistently recovers sharp images and accurate kernels.

## 2 RELATED WORK

Blind image deblurring has been extensively studied under probabilistic frameworks, with Variational Bayes (VB) methods being a prominent approach (Fergus et al., 2006). VB-based techniques model both the latent sharp image and the blur kernel as random variables and estimate their posterior distributions, often incorporating priors for image gradients. A drawback is that they can be more sensitive to initialization and may converge to local minima, limiting their performance in challenging scenario. Deep Image Prior (DIP) methods have also been explored, where a randomly-initialized convolutional network is optimized to fit the blurred image, using the network's implicit structure as a regularizer (Ulyanov et al., 2018). These methods can be prone to overfitting to noise and artifacts if not properly regularized or early-stopped.

In recent years, deep learning techniques have emerged as powerful alternatives to traditional VB and MAP-based blind deblurring methods. Unlike classical approaches that rely on hand-crafted priors, deep networks are capable of learning complex image statistics and blur distributions directly from large-scale datasets. Early works, such as multi-scale convolutional networks proposed by Nah et al. (2017), aim to directly restore sharp images from blurry inputs in an end-to-end fashion, bypassing explicit kernel estimation. These methods leverage hierarchical features to handle spatially varying blur and have demonstrated significant improvements over conventional approaches in both qualitative and quantitative evaluations. Another line of research focuses on explicitly predicting the blur kernel using neural networks. Sun et al. (2015) proposed a patch-wise kernel prediction strategy, where the network estimates local blur kernels that are subsequently used for non-blind deconvolution. More recent works explore unsupervised or self-supervised learning schemes, incorporating cycle-consistency or reconstruction losses to mitigate domain gaps between synthetic and real-world data Kupyn et al. (2018); Tao et al. (2018).

Recently, diffusion models have emerged as powerful generative priors for solving blind image deblurring. (Murata et al., 2023) proposed GibbsDDRM, a partially collapsed Gibbs sampler that alternates between diffusion-based image restoration and analytical updates of the degradation parameters. (Whang et al., 2022) introduced a stochastic refinement strategy that leverages diffusion sampling to iteratively improve deblurring quality. (Ren et al., 2023) further extended this line of work with a multiscale, structure-guided diffusion model that improves convergence and preserves fine details. More recently, (Laroche et al., 2024) combined diffusion models with an Expectation–Maximization framework, where the E-step samples the latent sharp image distribution using a diffusion prior and the M-step updates the blur kernel via a MAP estimation. Overall, deep learning approaches have demonstrated remarkable performance in blind image deblurring by learning flexible priors for both latent images and blur kernels. However, their performance can still degrade when faced with severe blur, unseen motion patterns, or domain shifts between training and test data.

## 3 BLIND DEBLURRING VIA SELF-DIFFUSION

This work addresses blind image deblurring by extending the **self-diffusion** framework, a general, zero-shot approach for solving inverse problems Luo & Huang (2025). The original self-diffusion method was designed for non-blind problems where the degradation operator is known. Our key contribution is to adapt this powerful framework to the more challenging blind deconvolution setting, where the degradation operator—specifically the blur kernel—is also unknown. To achieve this, we introduce a novel process that jointly recovers the clean image and the blur kernel by optimizing two coupled, untrained neural networks within a self-contained reverse diffusion process.

### 3.1 SELF-DIFFUSION

Self-diffusion is a training-free paradigm designed to solve general linear inverse problems of the form $\mathcal{A}\mathbf{x}_{\text{true}} = \mathbf{y}$, where $\mathcal{A}$ is a known forward operator, $\mathbf{x}_{\text{true}}$ is the unknown solution, and $\mathbf{y}$ is the observation. It operates via an iterative reverse diffusion process that starts from pure Gaussian noise. At each step $t$, a noisy version of the current estimate $\mathbf{x}_t$ is created with

$$\hat{\mathbf{x}}_t = \mathbf{x}_t + \sigma_t \cdot \epsilon_t .$$

A single, randomly initialized network at the first time step—the denoiser $D_\theta$—is then optimized continuously by minimizing a data fidelity loss with respect to the original observation $\mathbf{y}$ for each time step $t$,

$$\mathcal{L}_t(\theta) = \|\mathcal{A}D_{\theta,t}(\hat{\mathbf{x}}_t) - \mathbf{y}\|_2^2 \, .$$

The effectiveness of this process stems from a principle known as noise-regulated spectral bias. The noise schedule $\sigma_t$ implicitly regularizes the optimization, forcing the network to first learn low-frequency components and progressively refine high-frequency details in a multi-scale manner.

## 3.2 Joint Image and Kernel Estimation

The forward model for blind image deblurring is given by

$$\mathbf{y} = \mathbf{x}_{\text{true}} \circledast \mathbf{k} + n \, ,$$

where the sharp image $\mathbf{x}_{\text{true}} \in \mathbb{R}^{H \times W \times C}$ and the blur kernel $\mathbf{k} \in \mathbb{R}^{K \times K}$ are both unknown. To adapt the self-diffusion framework to this blind setting, we must estimate both variables simultaneously. We achieve this by employing two dedicated, randomly initialized networks: an image denoiser $D_\theta$ to restore $\mathbf{x}_{\text{true}}$, and a kernel generator $G_\phi$ to produce $\mathbf{k}$. Our method simulates a reverse diffusion process over $T$ discrete time steps, starting with random noise for both the image estimate, $\mathbf{x}_T$, and the kernel estimate, $\mathbf{z}_T$. At each time step $t \in \{T, T-1, ..., 1\}$, the current estimates are perturbed with scheduled noise,

$$\hat{\mathbf{x}}_t = \mathbf{x}_t + \sigma_t \cdot \epsilon_x, \quad \text{and} \quad \hat{\mathbf{z}}_t = \mathbf{z}_t + \sigma'_t \cdot \epsilon_z,$$

where $\epsilon_x \sim \mathcal{N}(0, \mathbf{I})$ and $\epsilon_z \sim \mathcal{N}(0, \mathbf{I})$. The noise schedule is $\sigma_t = \sqrt{1 - \bar{\alpha}_t}$, where $\bar{\alpha}_t = \prod_{i=0}^{t}(1 - \beta_i)$ and $\beta_t = \beta_{\text{end}} + \frac{t}{T-1}(\beta_{\text{start}} - \beta_{\text{end}})$, and $\sigma'_t = \mu\sigma_t, \mu = 0.15$. While the standard self-diffusion loss relies solely on data fidelity, the joint estimation of $\mathbf{x}$ and $\mathbf{k}$ is a severely ill-posed problem that requires additional constraints. Therefore, we augment the loss with an $\ell_1$ term for the kernel. The networks are jointly optimized within an inner loop by minimizing the following objective:

$$\mathcal{L}_t(\theta, \phi) = \|(D_\theta(\hat{\mathbf{x}}_t) \circledast G_\phi(\hat{\mathbf{z}}_t)) - \mathbf{y}\|_2^2 + \lambda_k R(G_\phi(\hat{\mathbf{z}}_t))$$

After the inner optimization loop, the improved networks produce cleaner estimates for the next time step, $\mathbf{x}_{t-1} = D_\theta(\hat{\mathbf{x}}_t)$ and $\mathbf{z}_{t-1} = G_\phi(\hat{\mathbf{z}}_t)$, continuing the coarse-to-fine reconstruction inherent to the self-diffusion process. The detailed algorithm is presented in Algorithm 1.

---

**Algorithm 1** Blind Deblurring using self-diffusion (DeblurSDI)

---

**Require:** Blurry image $\mathbf{y}$, total steps $T$, inner iterations $S$, learning rate $\eta$, $\ell_1$ weights $\lambda_k$
 1: **Initialize:**
 2: Image estimate $\mathbf{x}_T \sim \mathcal{N}(0, \mathbf{I})$; $D_\theta$ with random weights $\theta$
 3: Kernel estimate $\mathbf{z}_T \sim \mathcal{N}(0, \mathbf{I})$; $G_\phi$ with random weights $\phi$
 4: Adam optimizer for $(\theta, \phi)$
 5: Noise schedule $\sigma_t$ for $t \in \{1, ..., T\}$
 6: **for** $t = T, T-1, ..., 1$ **do**
 7:     Sample noise $\epsilon_\mathbf{x}, \epsilon_\mathbf{z} \sim \mathcal{N}(0, \mathbf{I})$
 8:     Create noisy inputs: $\hat{\mathbf{x}}_t \leftarrow \mathbf{x}_t + \sigma_t \cdot \epsilon_\mathbf{x}$, and $\hat{\mathbf{z}}_t \leftarrow \mathbf{z}_t + \sigma'_t \cdot \epsilon_\mathbf{z}$
 9:     **for** $s = 1, ..., S$ **do**
10:         Generate kernel: $\mathbf{k} \leftarrow G_\phi(\hat{\mathbf{z}}_t)$
11:         Compute denoised image: $\mathbf{x}_t \leftarrow D_\theta(\hat{\mathbf{x}}_t)$
12:         Calculate loss: $\mathcal{L}(\theta, \phi) \leftarrow \|(\mathbf{x}_t \circledast \mathbf{k}) - \mathbf{y}\|_2^2 + \lambda_k R(\mathbf{k})$
13:         Update parameters via gradient descent: $(\theta, \phi) \leftarrow (\theta, \phi) - \eta\nabla_{(\theta,\phi)}\mathcal{L}(\theta, \phi)$
14:     **end for**
15:     Update image estimate: $\mathbf{x}_{t-1} \leftarrow D_\theta(\hat{\mathbf{x}}_t)$
16:     Update kernel latent code: $\mathbf{z}_{t-1} \leftarrow G_\phi(\hat{\mathbf{z}}_t)$
17: **end for**
18: **return** Reconstructed image $\mathbf{x} \leftarrow \mathbf{x}_0$ and final blur kernel $\mathbf{k} \leftarrow \mathbf{k}_0$

---

# 4 IMPLEMENTATION AND ANALYSIS

## 4.1 NETWORK ARCHITECTURE AND TRAINING

Due to the low-dimensional nature of the blur kernel, we employ a fully-connected network (FCN) to implement the kernel generator, $G_\phi$. To ensure the output corresponds to a physical blur kernel, a softmax activation is applied to the final layer, enforcing non-negativity and a sum-to-one constraint. The 1D output of $G_\phi$ is subsequently reshaped into a 2D blur kernel. Besides, we introduced standard mode for ablation study where the latent vector $z$ is sampled from a normal distribution and kept fixed during training, and diffusion mode where the $\mathbf{z}_t$ evolves through the self-diffusion process. $K$ is the kernel size, $n$ is the number of hidden layers and $H_d$ is the hidden dimension size. Figure 1 shows how the architecture of $G_\phi$ affects the performance of DeblurSDI. For the image

Table 1: The architecture of the kernel generator $G_\phi$

| Mode | Layer | Specification |
|------|-------|---------------|
| Standard | Input | $z \in \mathbb{R}^{200} \sim \mathcal{N}(0, \mathbf{I})$ |
| | Hidden layer | Linear(200, 2000); ReLU6 |
| | Output layer | Linear(2000, $K \times K$); Softmax |
| Diffusion | Input | $\mathbf{z}_t \in \mathbb{R}^{K \times K}$ |
| | Hidden layer 1 | Linear($K \times K, H_d$); ReLU |
| | Hidden layer 2 | Linear($H_d, H_d$); ReLU |
| | . . . | . . . |
| | Hidden layer $n$ | Linear($H_d, H_d$); ReLU |
| | Output layer | Linear($H_d, K \times K$); Softmax |

denoiser $D_\theta$, we employ an encoder-decoder network with skip connections, following a U-Net-like structure. The network consists of five hierarchical levels. Each level in the encoder path consists of two convolutional blocks and a stride-2 convolution for downsampling. Correspondingly, the decoder path uses bilinear upsampling. Skip connections concatenate features from each encoder level to the corresponding decoder level. Non-Local Blocks are integrated into the deeper encoder levels (levels 3, 4, and 5) to capture long-range dependencies. The architecture is detailed in Appendix B.

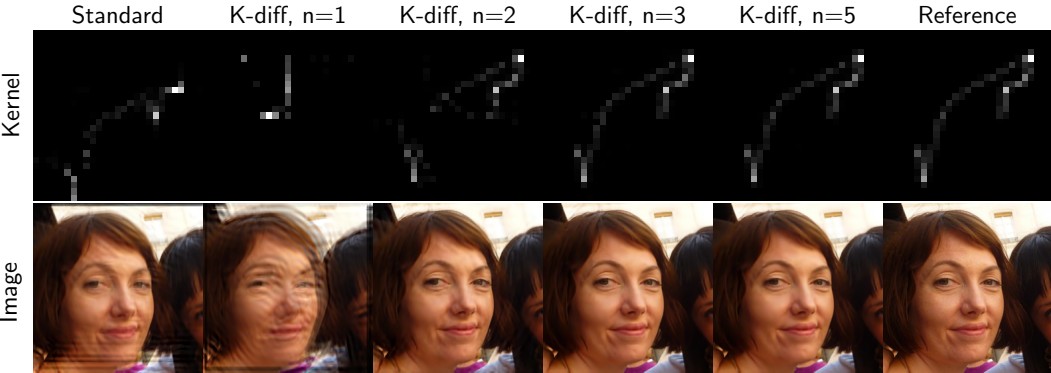

Figure 1: Ablaion study on the network for blur kernel estimation. This figure compares the performance of the "Standard" mode against the "Diffusion" mode architecture with a varying number of hidden layers ($n = 1$ to $n = 5$). The results clearly indicate that increasing the depth of the kernel generator network leads to a more accurate kernel estimation

We use a single Adam optimizer to jointly update the parameters of both the image denoiser, $D_\theta$, and the kernel generator, $G_\phi$. The initial learning rate for the image denoiser $D_\theta$ is set to $1 \times 10^{-3}$. The kernel generator $G_\phi$ uses a lower learning rate, typically 25% of the denoiser's rate (i.e., $2.5 \times 10^{-4}$), as the small change in the kernel can make a bigger impact on the image. The smaller learning rate helps stable convergence of the kernel estimate. Furthermore, we employ an optional adaptive learning rate schedule for the kernel generator. The learning rate is decayed by a factor of 0.95 at

the end of each outer time step $t$, down to a minimum threshold of $1 \times 10^{-5}$. The L1 regularization weight for the kernel prior is set to $\lambda_k = 2 \times 10^{-3}$. The noise level $\sigma_t$ for the image perturbation at each step $t$ is determined by a pre-defined variance schedule. Following common practice in diffusion models, we use a linear schedule where the variance $\beta_t$ interpolates from $\beta_{\text{start}} = 1 \times 10^{-4}$ to $\beta_{\text{end}} = 2 \times 10^{-2}$ over $T$ steps. The noise level $\sigma_t$ is then derived from the cumulative product of these variances.

## 4.2 HYPERPARAMETERS SENSITIVITY

The number of outer diffusion steps $T$ and inner optimization iterations $S$ are two critical hyperparameters that directly impact both the reconstruction quality and computational cost. Intuitively, more diffusion steps allow for a finer coarse-to-fine reconstruction, while more inner iterations enable better convergence of the networks at each step. However, increasing either parameter also leads to longer runtimes. To evaluate the sensitivity of our method to these parameters, we conduct experiments varying $T$ from 10 to 40 and $S$ from 25 to 500. As shown in Figure 2, performance improves with higher values of $T$ and $S$. The most significant gains occur when increasing $T$ from 10 to 30. However, the improvements tend to saturate beyond certain thresholds (e.g., $T = 30$, $S = 400$), with the performance curve for $T = 40$ closely tracking that of $T = 30$. This indicates that our approach can achieve strong deblurring performance without requiring excessively high iteration counts.

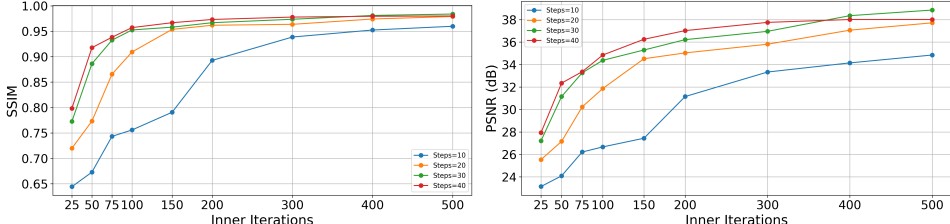

Figure 2: Sensitivity of noise steps, $T$, and inner iterations, $S$. The graphs show the SSIM (left) and PSNR (right) scores for different numbers of outer diffusion steps ($T \in \{10, 20, 30, 40\}$) and inner optimization iterations ($S \in \{25, ..., 500\}$).

## 4.3 DEBLURRING PROCESS

Figure 3 shows the evolution of estimates of image and kernel through the deblurring process. The left and right subfigures shows the SSIM and PSNR between the original image and the reconstruction over noise steps. The estimates of images and kernels at noise steps $5, 10, 15, 20, 30$ are shown on the top. Unlike traditional optimization processes where evaluation metrics typically increase monotonically, our curves exhibit an up-down-up behavior (especially for PSNR curve), which is attributed to the noise scheduling strategy. By injecting noise into intermediate reconstruction results, we effectively enlarge the search space of the inverse solution. The initial reconstructions are smooth and lack fine detail, while later steps recover sharper features. This aligns with the coarse-to-fine nature of self-diffusion.

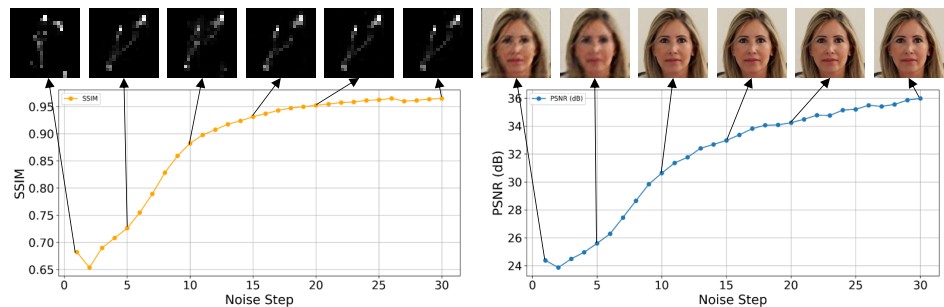

Figure 3: Evolution of image and kernel estimates during DeblurSDI's reverse diffusion process.

## 4.4 ROBUSTNESS TO KERNEL SIZE

The joint estimation of the image denoiser and the blur kernel often collapses to trivial solutions, such as Dirac kernels or reproducing the blurred image itself, especially when the chosen kernel size is incompatible with the image content. Larger kernels are harder to be recovered accurately, while smaller kernels may fail to capture long-range motion. For this reason, SelfDeblur (Ren et al., 2020) carefully selects the kernel size for each image. In contrast, our method exhibits much greater robustness. As shown in Figure 4, we evaluate ten different kernel sizes from 15 to 33, and compare the performance of several approaches. Our method not only achieves consistently superior performance across all kernel sizes but also demonstrates remarkable stability.

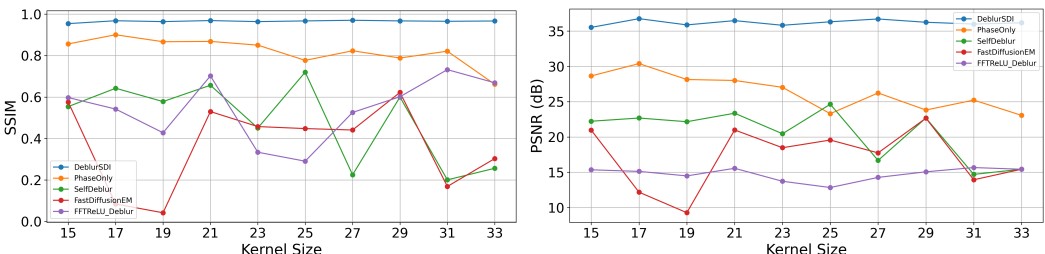

Figure 4: Performance and stability comparison across different kernel sizes. The graphs show the SSIM (left) and PSNR (right) scores for five deblurring methods evaluated on kernel sizes ranging from 15 to 33. Our method, DeblurSDI (blue), consistently achieves the highest scores and demonstrates remarkable stability, with its performance remaining largely unaffected by changes in kernel size. In contrast, other methods exhibit significant volatility, underscoring the superior robustness of our approach.

## 5 EVALUATION

### 5.1 DATASET

To systematically evaluate the performance of our method, we collected four datasets including Levin (Levin et al., 2007), Cho (Cho & Lee, 2009), Kohler (Köhler et al., 2012), and FFHQ (Karras et al., 2019). The first three datasets are widely used benchmarks for blind image deblurring, containing various synthetic blur kernels applied to natural images. The FFHQ dataset consists of 20 random selected human face images and 4 blur kernels from Cho (Cho & Lee, 2009). For our experiments, we use $T = 30$ outer reverse diffusion steps. At each step, we run $S = 200$ inner optimization iterations. For each dataset, we use the provided blur kernels and generate blurred images by convolving the sharp images with these kernels to simulate real-world conditions. Each blur kernel is applied to every image in the dataset. The details of our comprehensive datasets are shown in Table 2.

Table 2: Details of datasets for evaluation

|  | Image size | Kernel Size | Pairs |
|---|---|---|---|
| Levin[1] | 255×255 | (19, 17, 15, 27, 13, 21, 23, 23) | 32 |
| Cho[2] | 622×463, 780×580, 1006×665, 1002×661 | (27, 23, 19, 21) | 16 |
| Kohler[3] | 800×800 | (16, 14, 9, 13, 29, 17, 19, 98, 102, 62, 40, 29) | 36 |
| FFHQ[4] | 256×256 | (27, 23, 19, 21) | 80 |
| **Total** | - | - | 128 |

[1](Levin et al., 2007) [2](Cho & Lee, 2009) [3](Köhler et al., 2012) [4](Karras et al., 2019)

### 5.2 COMPARISON WITH OTHER METHODS

We compare our DeblurSDI method with several other blind deblurring approaches, including Phase-Only (Pan et al., 2019), FFT-ReLU Deblur (Al Radi et al., 2025), SelfDeblur (Ren et al., 2020), and FastDiffusionEM (Laroche et al., 2024). We evaluate the performance of each method

using Peak Signal-to-Noise Ratio (PSNR) and Structural Similarity Index Measure (SSIM) on the four datasets mentioned above. The quantitative results are summarized in Table 3. Rather than setting specific kernel size for each image (Ren et al., 2020), we set a fixed kernel size for each dataset, i.e., 27 for Levin, 33 for Cho, Kohler and FFHQ. As shown in the table, our DeblurSDI consistently outperforms all compared methods across all datasets, achieving significant improvements in both PSNR and SSIM metrics. This demonstrates the effectiveness and generalizability of our self-diffusion approach in recovering sharp images and accurate blur kernels.

Table 3: Quantitative results of blind deblurring performance (PSNR/SSIM) on four datasets.

|  | Phase-Only[1] | FFT-ReLU Deblur[2] | SelfDeblur[3] | FastDiffusionEM[4] | DeblurSDI (Ours) |
|---|---|---|---|---|---|
| Levin[a] | 20.68/0.6061 | 15.56/0.3845 | 25.06/0.7301 | 16.55/0.4005 | 31.85/0.7911 |
| Cho[b] | 19.89/0.6746 | 18.73/0.6546 | 20.37/0.6844 | 15.39/0.4687 | 28.73/0.8859 |
| Kohler[c] | 28.23/0.8092 | 25.33/0.7140 | 21.97/0.5995 | 18.85/0.4813 | 29.17/0.7653 |
| FFHQ[d] | 25.80/0.7904 | 21.71/0.6579 | 19.82/0.5563 | 15.59/0.3592 | 33.90/0.9064 |

[1] (Pan et al., 2019), [2] (Al Radi et al., 2025), [3] (Ren et al., 2020), [4] (Laroche et al., 2024)
[a] (Levin et al., 2007), [b] (Cho & Lee, 2009), [c] (Köhler et al., 2012), [d] (Karras et al., 2019)

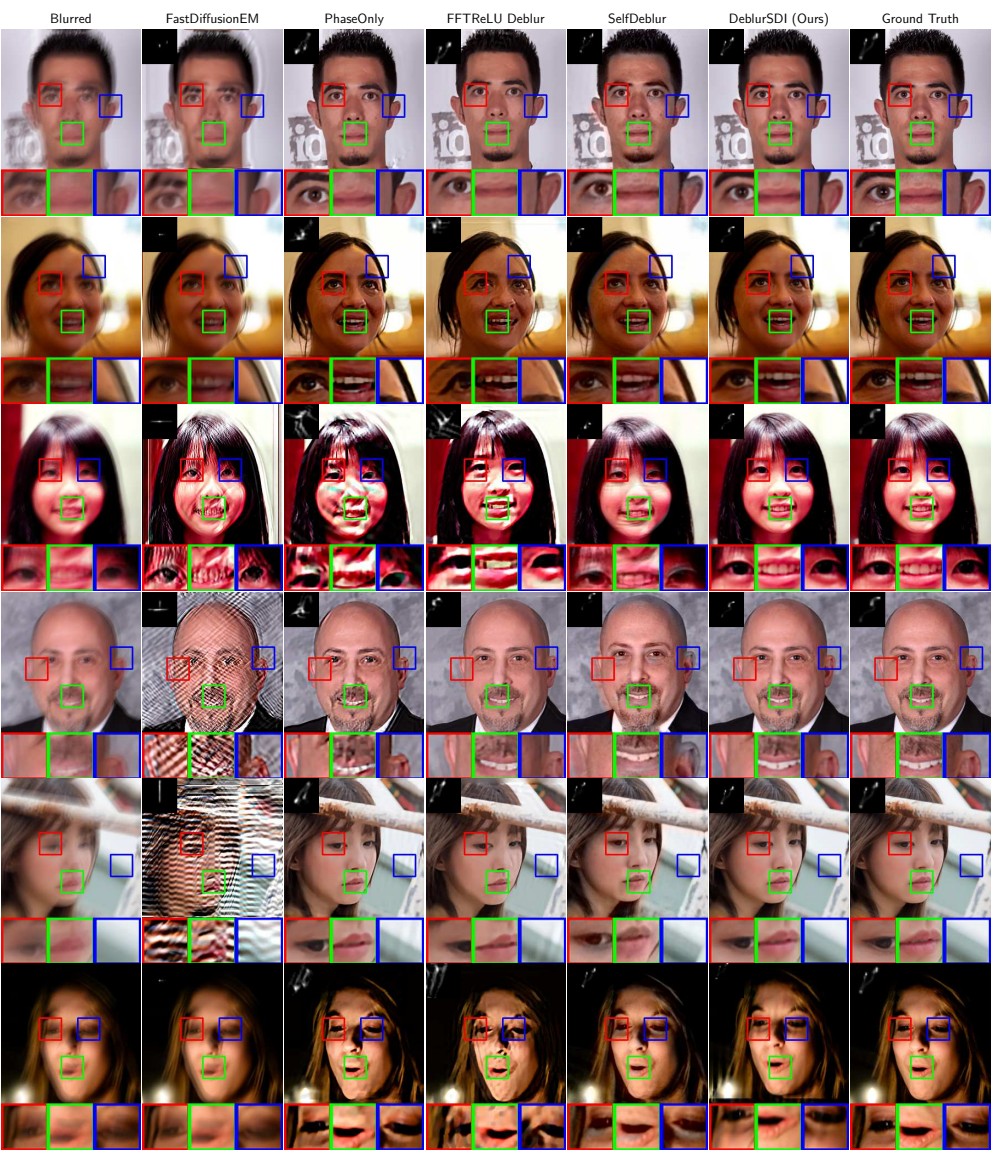

Figure 5: Deblurring results on the FFHQ dataset (Karras et al., 2019).

Figure 5 shows deblurring results of different methods on the FFHQ dataset (Karras et al., 2019). For each estimated image, the recovered kernel is displayed at the top-left corner, and three zoomed-in regions highlight fine details. As observed, FastDiffusionEM (Laroche et al., 2024) performs the worst. Despite being pre-trained on the FFHQ dataset, its performance remains poor: the estimated kernels degenerate into trivial point- or line-like structures. Without reliable kernel estimation, even a strong image prior cannot yield satisfactory deblurring. In contrast, other four optimization-based methods produce visibly superior results. PhaseOnly (Pan et al., 2019) and FFTReLU Deblur (Al Radi et al., 2025) provide good results in some cases, but lack of robustness when facing various blur kernels. SelfDeblur (Ren et al., 2020) shows the most promising performance out of previous methods, yet still suffers from poor generalizability, reconstruction shiftting and color distortion. The proposed self-diffusion-based DeblurSDI method consistently outperforms all compared methods, achieving significant improvements in the long-standing shiftting and robustness issues of blind deblurring techniques. Moreover, DeblurSDI is also able to recover accurate blur kernels, which gurantees the performance of the method. More results of other three datasets can be found in Appendix C.

## 6 CONCLUSION

This paper presents a novel self-diffusion-based approach for blind image deblurring, which we call DeblurSDI. Our method leverages the self-diffusion principle to recover accurate blur kernels and sharp images in a single framework. Experimental results on four benchmark datasets show that DeblurSDI consistently outperforms other blind deblurring methods on various datasets, demonstrating its effectiveness and generalizability in recovering sharp images and accurate blur kernels.

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

## A    LLM USAGE STATEMENT

We used a large language models only to refine grammar and improve the clarity of language in this manuscript. No part of the research ideation, experiment design, or analysis was performed by an LLM.

## B    ARCHITECTURE OF THE IMAGE DENOISER ($D_\theta$)

Table 4: The architecture of the image denoiser $D_\theta$, consisting of five encoder units ($e_i$) and five decoder units ($d_i$). The form is Conv(input channels, output channels, kernel size).

| Layer | Specification |
|---|---|
| **Input** | Noisy image $\hat{x}_t \in \mathbb{R}^{C \times H \times W}$ |
| **Output** | Denoised image $x_{t-1} \in \mathbb{R}^{C \times H \times W}$ |
| Encoder unit 1 | $e_1(\cdot, 128, 3)$ |
| Encoder unit 2 | $e_2(128, 128, 3)$ |
| Encoder unit 3 | $e_3(128, 128, 3)$ |
| Encoder unit 4 | $e_4(128, 128, 3)$ |
| Encoder unit 5 | $e_5(128, 128, 3)$ |
| Decoder unit 5 | $d_5(128, 128, 3)$ |
| Decoder unit 4 | $d_4(128, 128, 3)$ |
| Decoder unit 3 | $d_3(128, 128, 3)$ |
| Decoder unit 2 | $d_2(128, 128, 3)$ |
| Decoder unit 1 | $d_1(128, 128, 3)$ |
| Output layer | Conv(128, C, 1); Sigmoid |

## C    MORE RESULTS

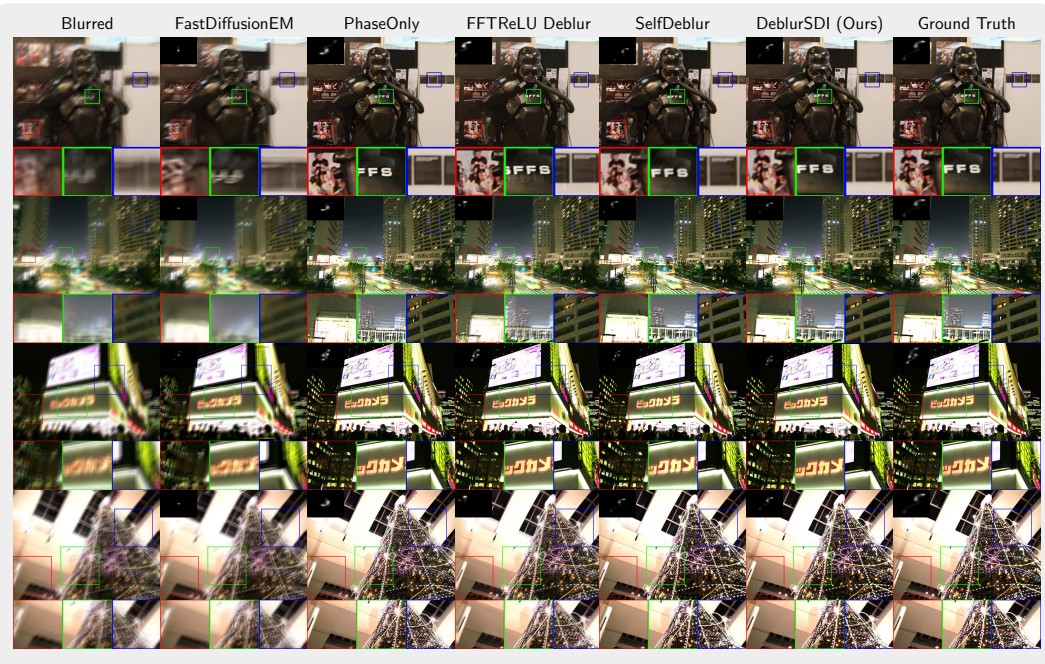

Figure 6: Deblurring results on the Cho dataset (Cho & Lee, 2009).

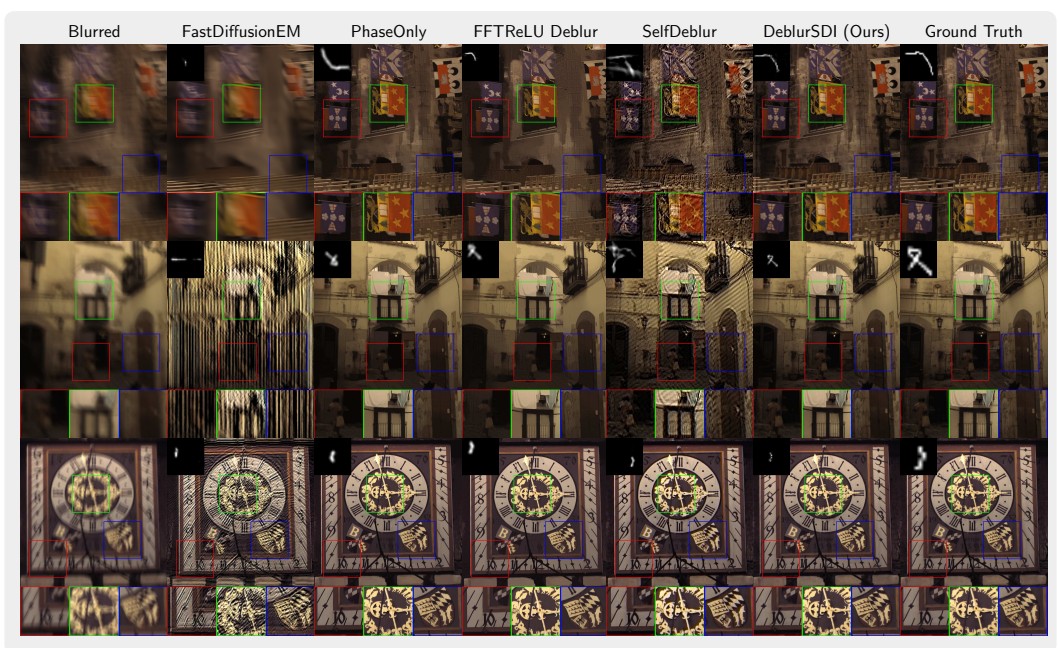

Figure 7: Deblurring results on the Kohler dataset (Köhler et al., 2012).

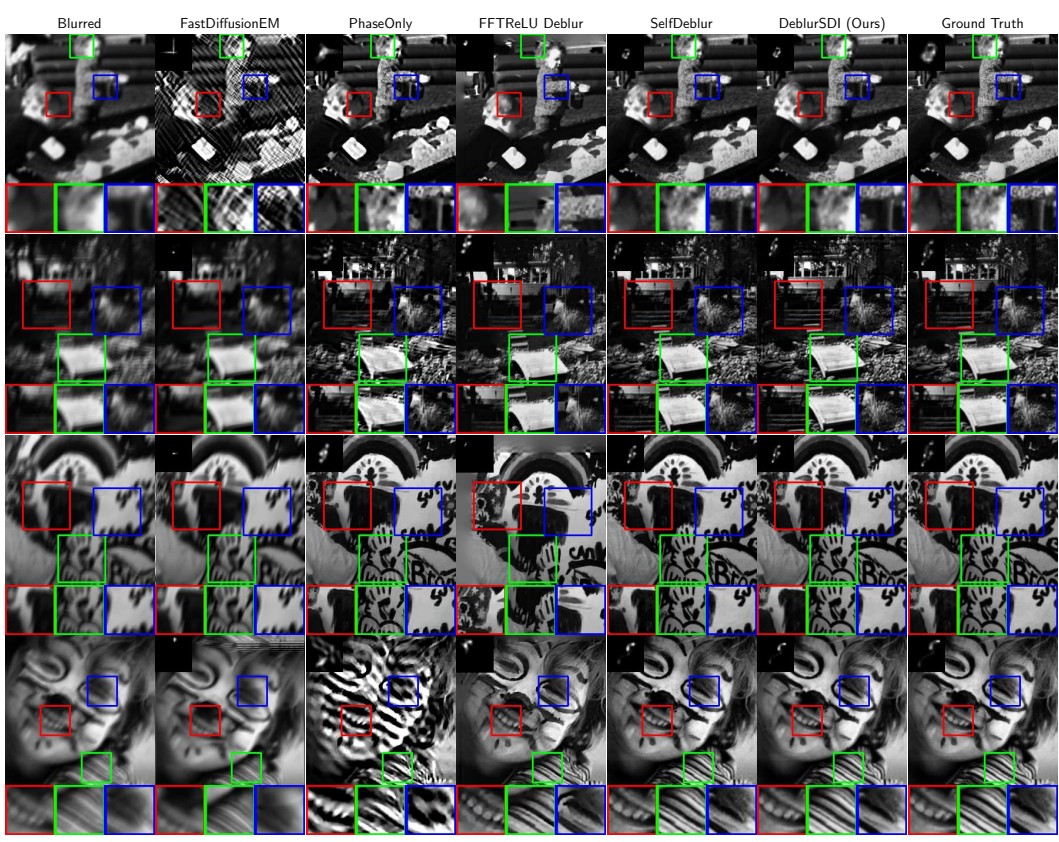

Figure 8: Deblurring results on the Levin dataset (Levin et al., 2007).

