# OpenReview forum: "DeblurSDI: Blind Image Deblurring Using Self-diffusion"
_ICLR.cc/2026/Conference — ICLR 2026 Conference Withdrawn Submission_

### Official Review · Reviewer_Em1c · 2025-10-19

**Soundness:** 2
**Presentation:** 2
**Contribution:** 2
**Rating:** 4
**Confidence:** 4

**Summary:**

This paper proposes DeblurSDI, a zero-shot, self-supervised blind deblurring framework built upon a self-diffusion process. The method jointly optimizes two randomly initialized networks — an image denoiser and a blur-kernel generator — within a reverse diffusion loop. A custom noise scheduler is introduced to stabilize optimization and improve robustness to kernel-size variations. Experiments on Levin, Cho, Köhler, and FFHQ datasets show that DeblurSDI achieves higher PSNR/SSIM than several baselines.

**Strengths:**

* The idea of integrating a multi-step noise schedule into an inner-loop, training-free reconstruction loop is practically interesting
* The framework is cleanly described and easy to reproduce.

**Weaknesses:**

* The core idea is essentially: take SelfDeblur/DIP-style joint optimization and run it under a reverse-noise-schedule / diffusion loop. This is an architectural substitution rather than a conceptual advance. Similar themes exist in DIP/SelfDeblur and in diffusion-based blind inverse solvers (EM-with-diffusion, GibbsDDRM, stochastic refinement). The manuscript does not convincingly argue why diffusion (noise schedule) yields a fundamentally different regularization or identifiability property in the blind setting. The claimed “key innovation = noise scheduling” is not substantiated by either theory or rigorous diagnostics.

* The experimental comparison is insufficient and thus unconvincing. Only a limited set of baselines is considered, particularly within the SelfDeblur family of optimization-based methods. Discussions on more recent unsupervised or zero-shot approaches, e.g., [1], [2], should be included.

* The method section is too thin and lacks sufficient technical depth. The paper does not clearly explain how the proposed framework differs fundamentally from SelfDeblur. In addition, the related work section is incomplete, where important recent unsupervised approaches such as [1] and [2] are missing.

[1] Xiaole Tang et al., Uncertainty-aware unsupervised image deblurring with deep residual prior, CVPR 2023.

[2] Jiangtao Zhang et al., Blind image deconvolution by generative-based kernel prior and initializer via latent encoding, ECCV 2024.

**Questions:**

See weaknesses.

---

### Official Review · Reviewer_XQXi · 2025-10-26

**Soundness:** 1
**Presentation:** 1
**Contribution:** 2
**Rating:** 0
**Confidence:** 4

**Summary:**

Motivated by the deep image prior, which has shown that the inductive bias of an (untrained) deep neural network suffices as regularizer for image restoration, this paper proposes a training-free strategy for blind image deblurring. Based on a standard linear blur model with a spatially-fixed kernel of given size, the paper simultaneously estimates the restored image and the blur kernel with a self-diffusion framework. Specifically, the paper learns two denoising networks in a diffusion framework at test time that remove noise added to the image respectively kernel such that their convolution is close to the blurry input image. The blur kernel is regularized via an L1 sparsity prior. Experiments (with questionable design, see below) show that this outperforms other recent blind deblurring methods on several standard datasets.

**Strengths:**

Simultaneously estimating both the deblurred image and a spatially-invariant blur kernel using a self-diffusion framework is certainly an interesting and potentially promising approach to blind image deconvolution. The idea seems to make intuitive sense.

**Weaknesses:**

Very significant issues:
* The paper builds upon the previous self-diffusion work of Luo & Huang (2025). This paper is not published yet, which makes it impossible for the reviewer to judge the novelty of this submission over the cited previous work. The standard thing to do in such a case would have been to include a copy of a preprint of the unpublished previous work as supplemental material, but the authors have not done this. [NB. I am leaving aside the question whether this also de-anonymizes the ICLR submission as we cannot conclude with certainly that the authors of this submission are the same as that of the unpublished prior work.]
* Another significant difficulty is that this submission refers to several concepts from Luo & Huang (2025), such as the noise-regulated spectral bias (l. 113), which the reviewers cannot understand due to the unavailability of the prior work, yet are important to understand how the proposed approach works and why it works the way it does.
* The non-local blocks that are used in the image diffusion model (l. 189f) are neither explained, nor is any reference given, nor do they appear in Table 4. The architecture is highly unclear and the results are not reproducible.
* The experimental setup seems to be erroneous. L. 306f suggests that the blurry images are generated by convolving the clean image with the known kernel, yet there is no mention of noise being added after the fact. Without adding noise, the deconvolution problem becomes not only significantly easier, but it is also standard in the literature to add noise for experiments with synthetic blur. Not adding some noise in the generation process makes the comparison to previous methods potentially unfair (numbers may not be comparable).
* I have significant doubts about the experimental results. The numerical difference to previous methods (Figs. 4 and Table 3) is so high that it is really difficult to believe that these comparisons are fair. A difference of 10-15 dB is outside of the credible range and also not consistent with the visuals in Fig. 5. For example, the PSNR reported for FFHQ in Laroche et al. (2024) is 8 dB higher than the results reported for the same method here, yet there is no indication what might explain this difference.

Major points:
* The discussion of related work is quite sparse. There is other work in the literature that performs blind image restoration via the interplay of two diffusion processes, e.g. [A]. There are differences since [A] does not use self-diffusion but this paper should be cited and compared to anyway. Other recent relevant work includes [B], [C], [D].
* The ablation study is very limited, for example:
  - The noise schedule for the image ($\sigma_t$) is not explained or ablated.
  - The noise schedule for the blur kernel ($\sigma_t’$) is not justified or ablated. Why do we need a lower noise level for the blur kernel and where does the constant $0.15$ comes from?
  - There is no ablation study w.r.t. the regularization weight $\lambda_k$.
  - The number of hidden layers $n$ is ablated only by showing visual results (Fig. 1) without any numbers.
* The noise regularizer R is not formally defined. Only from the text we can infer that an $\ell_1$ term is used.
* It is not very clear what the standard mode in l. 170 is. Why are there fixed parameters for the standard mode (top of Table 1), whereas for the kernel diffusion model (bottom of Table 1) the parameters are variable. I also could not find how $H_d$ is chosen.
* The experimental analysis in Fig. 2 and 3 is unclear. Fig. 2 seems to suggest that SSIM and PSNR of the image are being evaluated. On the other hand, Fig. 3 seems to suggest that SSIM pertains to the kernel (left) and PSNR to the image (right). Why is this? Shouldn’t both be evaluated? Where and how is the kernel estimation error quantified (cf. Laroche et al., 2024 and many others).

Minor points:
* The approach is limited to spatially-invariant blur. This is fine, in principle, but should be clearly stated. The introduction, for example l. 032f, suggest something more general than the paper later fulfills (cf. l. 120ff).
* The citation style is not applied correctly throughout the paper.
* There are no equation numbers.

[A] Hyungjin Chung, Jeongsol Kim, Sehui Kim, Jong Chul Ye, “Parallel Diffusion Models of Operator and Image for Blind Inverse Problems”, CVPR 2023
[B] Yash Sanghvi, Yiheng Chi, Stanley H. Chan, “Kernel Diffusion: An Alternate Approach to Blind Deconvolution”, ECCV 2024
[C] Hamadi Chihaoui, Abdelhak Lemkhenter, Paolo Favaro, “Blind Image Restoration via Fast Diffusion Inversion”, NeurIPS 2024
[D] Hamadi Chihaoui, Paolo Favaro, “Diffusion Image Prior”, ICCV 2025

**Questions:**

The majority of issues cannot be resolved as part of the rebuttal / reviewer discussion such as Luo & Huang (2025) not being publicly available. While I highly doubt that a rebuttal will change my mind, here are some questions that would at the very least need to be resolved:
* Can you explain and give an intuition behind how the method works (e.g., noise-regulated spectral bias) and explain in detail how it differs from Luo & Huang (2025).
* Can you explain and justify the experimental methodology in more detail? Have the blurred images been generated correctly (i.e. with noise added)? Why do the results differ so much from previous work and the numbers reported there?
* Can you describe the architecture more clearly (see weaknesses above)?
* Is the method stable w.r.t. the parameters not analyzed in the paper (see weaknesses).
* How accurate is the method in terms of kernel estimation?
* The paper claims robustness to kernel size (e.g., l. 053), but what about robustness to kernel shape? Can anything be said about how well different types of blur kernels (e.g. Gauss vs. elongated blurs) can be recovered?
* l. 144: Does weight*s* imply that there is more than one weight? What do the other weights do?

---

### Official Review · Reviewer_8WxE · 2025-10-29

**Soundness:** 3
**Presentation:** 3
**Contribution:** 2
**Rating:** 6
**Confidence:** 3

**Summary:**

## Summary

This paper introduces DeblurSDI, a novel blind image deblurring framework based on self-diffusion principles, requiring no pre-training. The method formulates blind deconvolution as an iterative reverse self-diffusion process, starting from pure noise and progressively refining to recover a sharp image and blur kernel. A key innovation is the simultaneous and continuous optimization of two randomly initialized neural networks: one for generating the sharp image and the other for the blur kernel. By introducing a novel noise scheduling mechanism, the method stabilizes the joint optimization process and exhibits remarkable robustness to variations in blur kernel size. Experimental results demonstrate that DeblurSDI consistently achieves superior performance on multiple benchmark datasets, outperforming current state-of-the-art methods in both recovering sharp images and accurately estimating blur kernels.

**Strengths:**

## Strengths

The most significant strength of this research is its zero-shot and self-supervised nature, eliminating the reliance on large external datasets for pre-training. This drastically improves the model's generalization ability and adaptability to unknown blur types and real-world scenarios. Secondly, the proposed noise scheduling mechanism effectively addresses the instability issues in jointly estimating the image and blur kernel, enabling the method to be highly robust to the choice of blur kernel size, overcoming a crucial limitation of many existing methods. Finally, extensive experiments and detailed comparative analyses show that DeblurSDI achieves state-of-the-art performance on several challenging benchmark datasets, comprehensively surpassing existing methods in both quantitative metrics (PSNR/SSIM) and visual effects.

**Weaknesses:**

## Weaknesses

Despite its excellent performance, there are some potential drawbacks. Firstly, the computational cost of this method may be high. As an iterative optimization process, it requires performing multiple (e.g., S=200 times) internal optimization loops at each diffusion step, which can lead to long processing times for a single image, limiting its potential in applications requiring real-time processing.

Secondly, the performance of this method relies on the selection of a series of hyperparameters, such as the number of outer diffusion steps T, the number of inner iterations S, the learning rate, and the L1 regularization weight. Although the paper conducts sensitivity analysis, finding the optimal combination of hyperparameters for images with different characteristics may still be a challenge in practical applications.

Finally, the experiments in this paper are mainly based on synthetic blur datasets. Although these are standard benchmark tests, the performance of this method on real-world blurred images caused by more complex nonlinear and spatially varying blurs remains to be further verified.

**Questions:**

## Questions

Building upon the potential weaknesses, several critical questions emerge regarding the practical deployment and robustness of DeblurSDI. Could the authors provide a detailed analysis of the computational overhead, including the typical inference time per image, and benchmark it against pre-trained methods that require only a single forward pass? Have any strategies been considered to accelerate the iterative optimization process to make it viable for time-sensitive applications like video deblurring? Furthermore, regarding hyperparameters, what was the methodology for selecting the reported values for T and S, and how significant is the performance degradation with suboptimal settings? Finally, to truly assess its generalizability, how does the model perform on real-world blurred images that feature more complex, non-linear, or spatially-varying degradations, which are not fully captured by the synthetic datasets used in the evaluation?

---

### Official Review · Reviewer_edWv · 2025-11-01

**Soundness:** 2
**Presentation:** 1
**Contribution:** 2
**Rating:** 2
**Confidence:** 3

**Summary:**

The paper proposes a new image deblurring algorithm, called DeblurSDL to estimate both the true image and the blur kernel using a self-diffusion process originated from a 2025 paper by Luo and Huang. The only difference appears to be a new kernel generator network in the fidelity term to estimate the blur kernel, and an L1-norm on the blur kernel for regularization. Results in Table 3 show significant gain over pervious deblurring methods.

**Strengths:**

The (limited) results show in Table 3 appear to be fantastic, significantly outperforming previous blind deblurring methods.

**Weaknesses:**

1. The writeup appears to be sub-par. None of the equations are properly numbered. Symbols are not properly defined; for example, in the equation at the bottom of page 2 (not numbered), what are $\sigma_t$ and $\epsilon_t$? There are also grammatical / spelling errors, such as "ill-pose" in page 1, and "Ablaion study" in the caption of Figure 1.

2. The differentiation from the original self-diffusion paper by Luo & Huang 2025 appears to be minimal. The only discernable differences appear to be a new black box $G_{\phi}(\cdot)$ in the fidelity term and an added $\ell_1$-norm regularization term for the blur kernel.

3. The first two equations in page 3 (neither of them is properly numbered) appear to be inconsistent. The second equation states that the true image x is first blurred, then noise is added. If this is the forward model, the corresponding fidelity term should be $\| D_{\theta}(\mathcal{A} \mathbf{x}) - \mathbf{y} \|^2_2$, not $\|\mathcal{A} D_{\theta}(\mathbf{x}) - \mathbf{y}\|^2_2$. I understand that the first equation is used in a reverse diffusion step. However, it still does not sufficiently explain the following objective in the third equation in the page (also not properly numbered).

4. It is not clear why an $\ell_1$-norm for the blur kernel is sufficient to regularize the severely ill-posed blind deblurring problem. Why is the $\ell_1$-norm better than the $\ell_2$-norm (commonly used)? Why are there no regularization terms for the image itself?

5. It is not clear what is novel about the noise scheduler(s), which appear to be standard in a diffusion model.

6. The number of comparison schemes in Table 3 is too small. For example, there is no comparison with the following image deblurring scheme that is also based on diffusion:

X. Li, Y. Chi, and S. H. Chan, “Kernel Diffusion: An Alternate Approach to Blind Deconvolution,” in Proceedings of the European Conference on Computer Vision (ECCV), Milan, Italy, October 2024, pp. 317–334.

**Questions:**

1. What are the key "non-trivial" differences between this paper and the original self-diffusion paper by Lou & Huang 2025? There are deblurring results in that paper already; see box (b) in Fig.1 of the original paper.

2. Why $\ell_1$-norm of the blur kernel is sufficient to regularize the severely ill-posed blind image deblurring problem? Previous works typically have regularization terms for both the image and the blur kernel separately. Further, $\ell_2$-norm of the blur kernel has been previously used.

3. How would the proposal compare with other recent diffusion-based schemes such as the following? In particular, the following paper argues that alternately optimizing the image and the blur kernel would mean the solution is trapped in one of many poorly performing local minima, yet this is exactly what the current proposal is doing by designing two separate networks to optimize the two unknowns (image and blur kernel).

X. Li, Y. Chi, and S. H. Chan, “Kernel Diffusion: An Alternate Approach to Blind Deconvolution,” in Proceedings of the European Conference on Computer Vision (ECCV), Milan, Italy, October 2024, pp. 317–334.

**Details Of Ethics Concerns:**

I worked on image deblurring previously but have not kept up with the recent literature, but the results in Table 3 (more than 5dB gain in PSNR) appear too good to be true for the exceedingly simple scheme that the authors are proposing. I could be wrong, but in my professional opinion, I do not think the results are believable.

---

### Note · Authors · 2025-11-12

I have read and agree with the venue's withdrawal policy on behalf of myself and my co-authors.